# Biomechanics of Ex Vivo-Generated Red Blood Cells Investigated by Optical Tweezers and Digital Holographic Microscopy

**DOI:** 10.3390/cells10030552

**Published:** 2021-03-04

**Authors:** Claudia Bernecker, Maria Augusta R. B. F. Lima, Catalin D. Ciubotaru, Peter Schlenke, Isabel Dorn, Dan Cojoc

**Affiliations:** 1Clinical Department of Blood Group Serology and Transfusion Medicine, Medical University of Graz, 8036 Graz, Austria; peter.schlenke@medunigraz.at (P.S.); isabel.dorn@medunigraz.at (I.D.); 2CNR-IOM, National Research Council of Italy—Institute of Materials, Area Science Park, 34149 Trieste, Italy; lima@iom.cnr.it (M.A.R.B.F.L.); ciubotaru@iom.cnr.it (C.D.C.); 3Physics Department, University of Trieste, 34127 Trieste, Italy

**Keywords:** red blood cells, optical tweezers, digital holographic microscopy, maturation, deformability

## Abstract

Ex vivo-generated red blood cells are a promising resource for future safe blood products, manufactured independently of voluntary blood donations. The physiological process of terminal maturation from spheroid reticulocytes to biconcave erythrocytes has not been accomplished yet. A better biomechanical characterization of cultured red blood cells (cRBCs) will be of utmost interest for manufacturer approval and therapeutic application. Here, we introduce a novel optical tweezer (OT) approach to measure the deformation and elasticity of single cells trapped away from the coverslip. To investigate membrane properties dependent on membrane lipid content, two culture conditions of cRBCs were investigated, cRBC^Plasma^ with plasma and cRBC^HPL^ supplemented with human platelet lysate. Biomechanical characterization of cells under optical forces proves the similar features of native RBCs and cRBC^HPL^, and different characteristics for cRBC^Plasma^. To confirm these results, we also applied a second technique, digital holographic microscopy (DHM), for cells laid on the surface. OT and DHM provided related results in terms of cell deformation and membrane fluctuations, allowing a reliable discrimination between cultured and native red blood cells. The two techniques are compared and discussed in terms of application and complementarity.

## 1. Introduction

The availability, quality and safety of blood products is a growing issue worldwide and the emerging supply bottleneck is hardly manageable with voluntary blood donations alone [1]. There is a worldwide effort to generate ex vivo-cultured red blood cells (cRBCs) from different sources [2,3,4], primarily for very specific therapeutic approaches in highly immunized patients and subsequently for broad clinical application by using a near universal RBC phenotype. However, prior to clinical use, there are some hurdles. In vivo, the terminal maturation comprises cytoskeletal remodeling from enucleated lobular reticulocytes to biconcave erythrocytes, but ex vivo, the triggers have remained elusive to date [1]. Reticulocytes retain residual RNA but already possess the ability to fulfill oxygen transport. Only after the terminal shape change to biconcavity do the mature erythrocytes achieve the crucial flexibility and stability within the blood stream [5]. This is necessary to adapt to various osmotic conditions as, while RBCs are biconcave disks at equilibrium, they shrink under hyperosmotic conditions and swell under hyposmotic conditions. During their lifespan of about 120 days, RBCs have to pass repeatedly through the microcapillaries and endothelial slits of the spleen, where morphology and functional integrity are verified, and malfunctioning RBCs are wiped out [6]. Healthy cells can stand these extreme deformations and keep their structural integrity. RBCs consist mainly of hemoglobin within the viscous cytoplasm, held together by a 2D cytoskeleton connected to a lipid bilayer. Their characteristic cellular properties are promoted by the cytoskeletal network of spektrin tetramers and actin-based junctional complexes, that withstand shear stress, and the lipid bilayer, which mainly supports the bending [7].

RBCs demonstrate a remarkable deformability when subjected to mechanical stresses [8]. This allows them to pass through capillaries with diameters smaller than the RBC diameter at rest and restore themselves to their original shape when they leave the capillaries. The deformability of RBCs plays an important role in their main function (transport of respiratory gases in the human body) and alterations in RBC deformability are related to several vascular diseases [9]. Therefore, cell deformability represents a natural choice to compare ex vivo-cultured red blood cells versus native RBCs.

During recent years, various experimental techniques have been developed to measure cell deformability and related biomechanical parameters for individual RBCs. They are based on micropipette aspiration, atomic force microscopy, optical tweezers (OTs), magnetic tweezers, digital holographic microscopy (DHM), dynamic light scattering, ektacytometry and microfluidics, thoroughly discussed in several reviews [10,11,12,13]. Since our study is based on OTs and DHM, hereafter, we focus only on aspects of these two techniques.

RBC bending by three optical traps [14], stretching RBCs by two counter propagating beams [15] and RBC buckling in single beam OTs [16,17] employed direct interaction of the cell with the radiation pressure of light in OTs. Although direct manipulation rapidly provides cell deformation in different ways, precise control of the forces is impossible. Applying calibrated forces can be achieved via silica microbeads bound to RBC membranes to study the elastic and viscoelastic behavior and measure the shear modulus [18,19]. A similar approach was used for detecting membrane fluctuation dynamics under controlled strain conditions [20]. Recently, the active nature of RBC flickering was investigated using active and passive microrheology of a single RBC in a sophisticated four-bead OT configuration [21]. The advances in OT technology, such as multiple and dynamic optical traps and circular to linear light polarization in OTs, allowed for multiple applications in studies of RBCs, as also shown in two recent reviews [22,23].

However, as for all single-cell measurement techniques, the typical time per experiment in OT assays can represent a limitation for some cell biology studies. Direct trapping and manipulation of RBCs by single beam OTs is certainly simpler and faster than indirect manipulation through attached beads. Although it does not provide a quantitative measurement of the shear modulus, direct manipulation allows a qualitative comparison of different biomechanical parameters between different types of RBCs.

Therefore, we decided to exploit single beam OTs to investigate the trapping and releasing times of the RBC from the trap, RBC buckling and folding and cell membrane fluctuations. An important characteristic of OTs is that the cell is kept in suspension by optical forces, avoiding possible influence from contact with neighboring surfaces. Interestingly, the distribution of the radiation pressure of light interacting with the RBC displays a clepsydra-like shape with the neck at the optical trap where the radiation pressure is the highest. This resembles RBC constriction through a capillary. To make the experimental scenario more like the physiological conditions, we implemented oscillatory OTs, which allowed us to displace the RBC through the fluid for a controlled number of cycles and velocities. Cell membrane fluctuations (CMFs) are monitored by recording the intensity fluctuations of the laser light scattered forward by the whole RBC in an optical trap. Since this signal provides an averaged CMF value obtained from different points over the whole cell, opposite from the point-oriented approaches previously used [24,25,26,27], we interrogate if this parameter can allow a proper discrimination between different types of cells.

To confirm the results obtained from OT experiments, we employed an additional optical technology: digital holographic microscopy (DHM). DHM, or quantitative phase imaging, is a valuable method for label-free investigation of cells and tissues in biomedicine, providing an objective measure of the 3D morphology and dynamics [28]. Evidence that membrane fluctuations in the RBC membrane have a metabolic as well as thermal energy component that are localized to the outer area of the cell has been shown by measuring cell membrane fluctuations in time-lapse DHM [29]. Another interesting application is the impact of storage on RBC stiffness by measuring the cell membrane fluctuations [30]. An important feature of DHM for RBC analysis is the possibility to extract relevant geometrical parameters of RBCs such as the cell volume and cell sphericity, and to determine the mean corpuscular hemoglobin [31]. Holograms of multiple cells settled on a coverslip can be recorded in parallel, increasing the throughput of the experiment. However, DHM requires additional time for numerical reconstruction of the holograms.

In this work, we present a comparative study on the mechanical properties of ex vivo-generated RBCs versus native RBCs using two complimentary approaches: OTs and DHM. Cell folding in an optical trap and cell membrane fluctuations of RBCs subjected to optical forces are explored by using OTs, while cell morphology and cell membrane fluctuations of RBCs settled on a planar surface are investigated by DHM. Remarkably, we found that both techniques provide a net discrimination between the two RBC types. These comprehensive investigations are of utmost importance to be aware of the overall quality and safety profile of cRBCs with regard to future in vivo applications in humans.

## 2. Materials and Methods

### 2.1. Cell Preparation

Human native erythrocytes (nRBCs) were obtained from fresh RBC units within 24 h. CD34^+^ cells were isolated from peripheral blood (purity 97.8 ± 0.7%) with the CD34 Microbead Ultrapure Kit (Miltenyi Biotec, Bergisch-Gladbach, Germany). Prior to sampling, written informed consent was obtained from volunteer donors. The study was approved by the local ethics committee of the Medical University of Graz, Austria in line with the Declaration of Helsinki (EK 27 165ex 14/15).

Erythroid differentiation from CD34^+^ hematopoietic stem/progenitor cells (HSPCs) was conducted according to the established three-phase culture model [32]. Iscove’s basal medium (Biochrome, Berlin, Germany) was supplemented with 5% human plasma (Octapharma, Vienna, Austria) (cRBC^Plasma^), or 2.5% human platelet lysate (cRBC^HPL^) (in-house production) from day 8 onwards. All media were supplemented with 10 µg/mL insulin (Sigma Aldrich, St. Louis, MO, USA) and 330 µg/mL human holo-transferrin (BBI solutions, Crumlin, UK). Cell differentiation was induced with 100 ng/mL stem cell factor (SCF), 5 ng/mL interleukin-3 (IL-3) (both PeproTech, Rocky Hill, NJ, USA) and 3 U/mL erythropoietin (EPO) (Erypo, Janssen Biologics B.V., Leiden, The Netherlands) and 10^−6^ M hydrocortisone (SigmaAldrich, St. Louis, MO, USA). Erythroid differentiation was monitored microscopically with May–Giemsa–Gruenwald (Hemafix, Biomed, Oberschleißheim, Germany) and neutral benzidine co-staining (o-dianisidine, SigmaAldrich, St. Louis, MO, USA). Additionally, the maturation stages were confirmed by flow cytometry (CD36, GPA (Beckman Coulter, Brea, California, USA); CD45, CD71 (Becton Dickinson, Franklin Lakes, NJ, USA)) on a CytoFLEX flow cytometer (Beckman Coulter, Brea, CA, USA). Dead cells were excluded by co-staining with 4,6-diamidino-2-phenylindole (DAPI; ThermoFisher, Waltham, MA, USA). After 18 days of differentiation, cells were filtered through a syringe filter (Acrodisc WBC syringe filter, Pall, Port Washington, NY, USA) to obtain the pure enucleated cRBC fraction free of precursors and expelled nuclei. Enucleated cRBCs were further characterized microscopically by new methylene blue staining for ribosomal residues (reticulocyte stain, Sigma-Aldrich, St. Louis, MO, USA) and flow cytometry for CD71 expression and on the basis of thiazole orange stain (ReticCount, Beckton Dickinson, Franklin Lakes, NJ, USA).

### 2.2. Cell Deformation and Cell Membrane Fluctuations in an Oscillatory Optical Trap

Cell deformation and cell membrane fluctuations of single erythrocytes confined in an optical trap were investigated using custom-built oscillatory optical tweezers (OOTs). The setup is schematically represented in Figure 1. It is based on the modular Thorlabs OT kit (OTKB, Thorlabs Inc., Newton, NJ, USA) with some modifications to allow probing cell biomechanics with silica microbeads [33,34]. To investigate cell deformation under optical forces, here, we trapped and manipulated the cell directly. The trapping laser (YLM-5, IPG Photonics GmbH, Oxford, Massachusetts, USA) is a continuous wave (CW) infrared (IR) fiber laser at 1064 nm and the optical trap can be moved axially in a range of ±8 µm from the laser focus, by using a focus tunable lens (FTL) (EL-10-30-NIR-LD, Optotune AG, Dietikon, Switzerland). The power of the IR laser at the sample was kept between 10 mW and 30 mW, allowing trapping without damaging. A high numerical aperture microscope lens (ML) is used for trapping and imaging (Nikon, NA 1.25, 100X, oil immersion, Nikon, Tokio, Japan) and a tube lens (f = 200 mm) for projecting the image on the Complementary Metal–Oxide–Semiconductor (CMOS) camera (DCU224M, Thorlabs Inc.). After cell identification, the trapping laser was switched on and the cell was picked up into the optical trap, positioned at a height H~20 µm from the coverslip. The cell was kept in the fixed optical trap for 10 s, then the trap was moved along the optical axis in an oscillatory movement with frequency f = 0.75 Hz and amplitude A = 5 µm (for 10 s), followed by a second oscillatory movement with higher amplitude A = 7 µm (for 10 s). Finally, by switching the laser off, the cell was released from the trap. Notice that using the FTL, the trapped cell can be shifted up and down without the need to change the position of the microscope objective. Cell trapping, folding and shape recovery were monitored with time-lapse microscopy, recording at 11 frames per second.

Cell membrane fluctuations are investigated using the interference pattern formed by the laser light scattered forward by the cell, collected by a microscope lens (Nikon, NA 0.3, 10X, not shown in Figure 1) and projected on a quadrant photo detector QPD (PDQ80A, Thorlabs Inc.) by a positive lens (f = 40 mm, Thorlabs Inc., not shown). The QPD allows for recording the intensity fluctuations of the light with a 5 kHz frequency bandwidth. The variance (σ^2^) and the power spectrum density (PSD) are calculated for the intensity fluctuation signal S(t) acquired by the QPD and given in mV:σ^2^ = <S^2^(t)>(1)
PSD(f) = FT{R_S_ (τ)}(2)
where R_S_(τ) = <S(t)∙S(t − τ)> is the autocorrelation function, FT is the Fourier transform and f is the frequency variable. The variance σ^2^ is the value of the autocorrelation function at delay τ = 0 and shows how much the signal values are spread from the mean value, while the PSD is a measure of the signal’s power content versus frequency. The signal S(t), the variance σ^2^ and the power spectrum density PSD are processed using custom Matlab code (Matlab R2017b, MathWorks, Natick, MA, USA).

### 2.3. 3D Cell Imaging with Digital Holographic Microscopy

The cell height can be obtained by numerical reconstruction of a hologram recorded on a digital camera. The setup to record the hologram is represented schematically in Figure 2. It is based on a Mach–Zender interferometer adapted to a custom inverted microscope [30,35]. The laser beam (520 nm, max 20 mW) from a diode laser (LP520-SF15, Thorlabs Inc.) is collimated and split into two beams: one passes through the RBC and one is used as a reference. The beam passing through the sample is again collimated and the two beams are then recombined to form an interference pattern (digital hologram) on the sensor of the CMOS camera (CS235MU, Thorlabs Inc). The power of the laser beam at the sample was less than 1 mW. The 3D profile of the RBC is obtained by numerically processing the digital hologram with custom Matlab algorithms (MathWorks.com). The CMOS sensor works at a 110 fps frame rate (exposure time: 9 ms), recording holographic movies from which cell membrane fluctuations are measured.

The height h at the point (x,y) of the RBC is calculated using the formula [30]:h(x,y) = λ∙OPD(x,y)/2π∙(n_c_ − n_m_)(3)
where λ is the wavelength of the light (λ = 520 nm), OPD(x,y) is the optical path difference obtained from hologram reconstruction, n_c_ is the refractive index of RBC (n_c_ = 1.418), n_m_ is the refractive index of the medium (n_m_ = 1.33). The refractive index n_c_ is considered the same for all the RBC types analyzed hereafter [30,35].

An example of a digital hologram and its reconstruction is shown in Figure 2b for a native RBC (nRBC). The diameter of the cell is about 7–8 µm, in agreement with the size measured with normal bright field microscopy. The additional information by DHM is the cell height, obtained with Equation (3). A height profile is taken along the red line and represented in Figure 2b, showing the characteristic profile for RBC height, with a dimple in the center. The max cell height is 2.3 µm, while the height at the dimple is 1.4 µm, giving a sphericity coefficient of CS = 1.4/2.3 = 0.61.

### 2.4. Calculation of the Cell Morphological Parameters and Cell Membrane Fluctuation

Using DHM, we measured morphological parameters and cell membrane fluctuations. The holographic reconstruction of the DHM of the cell from a single frame is enough to derive the morphological parameters:

1. Cell projected area, *CA*:(4)$$CA=N_{p}\cdot pa$$
where *N_p_* is the number of pixels inside the cell area and *pa* is the pixel area (for a square pixel of size *p* = 0.15 µm, *pa =* 0.0225 µm^2^).

2. Cell volume, *CV* is calculated by summing the volume corresponding to all the pixels within the cell area:(5)$$CV=pa\sum_{i}h_{i}$$

3. Cell equivalent height, *hm*:(6)$$hm=\frac{CV}{CA}$$
is the mean height of the cell assuming the cell was a cylinder with *CA*, the cell projected area.

4. Cell sphericity, *CS*:(7)$$CS=\frac{hD}{hR}$$
where *hR* is the cell height at the ring and *hD* is the height at the dimple (see the green and blue arrows in Figure 2). The cell sphericity is a shape parameter which helps to evaluate if the cell is biconcave or spherocyte-like. The ring region is the region where the cell height is maximum. The diameter of the ring is about half the diameter of the cell. In practice, we sample the ring for cell height for *n* = 8 points and the dimple for *n* = 3 points and we calculate *hR* and *hD* as the mean of these values.

5. The mean corpuscular hemoglobin, *MCH*:(8)$$MCH=\frac{10\cdot mPhase\cdot \lambda \cdot CA}{2\cdot \pi \cdot \alpha_{Hb}}$$
where *mPhase* is the mean phase value over the cell projected area *CA* and *α_Hb_* = 1.96 um^3^/pg is a constant known as a specific refraction increment related mainly to the protein concentration [31].

#### Cell Membrane Fluctuation (CMF)

To determine CMF, we calculated, for each pixel within the cell, the fluctuation of the cell height in time (t > 1 s at 110 fps) and the corresponding standard deviation for each pixel of the cell, *STD*_*pix_i_* [29,30]. The CMF value is calculated as the mean of *STD*_*pix* [31]:(9)$$CMF=1/N_{p}\sum_{i}STD\_pix_{i}$$

## 3. Results

For comparative OT and DHM analyses, cRBC^Plasma^ and cRBC^HPL^ were differentiated ex vivo from CD 34^+^ HSPCs over 18 days [32]. The cumulative expansion was on average 0.5 E05-fold and 0.4 E05-fold in cRBC^Plasma^ and cRBC^HPL^. Homogenous differentiation of the cells was monitored by morphology and flow cytometry analyses during cultivation. At the end of culture, >99% of all cultures expressed the erythroid marker glycophorine A (CD253a). CD36 expression decreased from >95% to 20%, which indicates terminal differentiation of the cells [36]. Purity of enucleated cRBC fractions was achieved for analyses by filtration on day 18 (98% purity). Flow cytometry analyses for CD71 expression revealed 90.2 % (ReticCount 58.4%) and 91.7% (ReticCount 36%) positivity for cRBC^Plasma^ and cRBC^HPL^, respectively. New methylene blue staining indicated a maturation grade of both cRBC^Plasma^ and cRBC^HPL^ between native reticulocytes and nRBCs according to their stained ribosomal rests.

### 3.1. Deformation of Native vs. Ex Vivo-Generated RBC under Optical Forces

We first investigated the deformation of nRBCs and cRBCs under optical forces, using the OOTs and manipulation approach described in the Materials and Methods. In a typical experiment, the cells were initially set on the surface of the glass coverslip and inspected by optical imaging. Once a single cell was individuated in the field of view of the microscope, the stage was moved to position the cell approximately under the optical trap (Figure 3, first column). Due to the interaction with the optical field, the cell was attracted by the optical forces to the center of the trap. During trapping, the optical forces induced cell buckling and then cell folding in the trap (Figure 3, second and third columns). The cell was kept for 30 s in the static and oscillatory trap, as described in the Materials and Methods, and then released from the trap (Figure 3, fourth and fifth columns) by switching off the laser. To evaluate the time required by the cell to recover from a folded shape to a discoid shape, the cell was monitored with time-lapse microscopy for other 30 s after release. The optical field generated around the center of the optical trap was the same for all the experiments. It looks like a symmetric clepsydra with the intensity of light increasing towards the neck [37]. When the cell interacts with this optical field, different forces apply at different points of the cell, causing a deformation/folding of the cell. Notice that for a defined symmetrical optical field, cell deformation depends only on the local cell geometry and its material properties. We observed that nRBCs have the same behavior, folding in the same manner in the optical trap (*n* = 22 experiments). The typical deformation consists in cell buckling and folding (Figure 3 and Appendix A). The folding time is about 0.5–1 s, in agreement with results previously reported for nRBCs [17]. After release, the cell recovered its shape in about 5–7 s. We repeated the same type of experiments with ex vivo-generated cRBC^HPL^ (*n* = 20) and observed that they folded in a similar way and recovered in the same range of time. A typical sequence is shown in Figure 3 (second row) and the trapping and release experiment in Appendix A. However, the ratio, Rf, between the diameter of the cell and the width of the folded cell is bigger for cRBC^HPL^ (Rf = 3.2 ± 0.6) than for the nRBCs (Rf = 2.2 ± 0.3), indicating that cRBCs fold more than nRBCs. Since the optical field is the same, this difference was attributed to the lower elastic energy of bending in cRBC^HPL^ than in nRBCs. This energy depends on the mean membrane curvature, which is related to the content of lipids, and on the cell morphology [16]. In fact, cRBC^HPL^ have larger diameters, but are thinner and flatter than nRBCs, making them easier to buckle and fold.

A different behavior from cRBC^HPL^ and nRBCs was observed for cRBC^Plasma^ (*n* = 21). Their initial shape was spherical and they were not deformed by the optical forces (Figure 3, third row and Appendix A). We attributed this to the spherical shape and a higher elastic energy, which could not be exceeded by the energy associated with the optical forces.

### 3.2. Cell Membrane Fluctuations of Native vs. Ex Vivo-Generated RBCs in OOTs

Cell membrane fluctuations (CMFs) were investigated by measuring the intensity fluctuations of the laser light scattered by the RBC in the trap, as described in the Materials and Methods. The intensity fluctuations, S(t), are due to two factors: (1) the displacement of the cell in the trap and (2) cell membrane flickering. Although these two components cannot be decoupled, the signal analysis might be useful if the goal is to compare the fluctuations for cells of a similar size, for which the contribution of the first component is assumed to be comparable, while the contribution of the cell membrane fluctuations is used to mark the difference between types of cells.

We acquired the signal S(t) for the three cell types during the trapping experiments described above. The QPD signal S(t) was sampled at 5 kHz. This is a large frequency bandwidth, thoroughly covering the frequency band of the cell fluctuations [21,24]. The variance σ^2^ of the signal S(t) was first calculated for each RBC in a fixed and oscillating trap and then the mean value and standard deviation of the variances for the same type of cells were determined. From the results obtained for the three types of RBCs, shown in Table 1, we found that the mean variances for cRBC^HPL^ (*n* = 30 cells) and nRBCs (*n* = 32 cells) are close and that both are higher than the mean variance for the cRBC^Plasma^ (*n* = 24 cells). Since a higher value of the variance is associated with higher CMFs, this result shows that the CMF of nRBCs is similar to that of cRBC^HPL^ and higher than that of cRBC^Plasma^. Since not all the distributions are normally distributed, we also performed a Mann–Whitney U test. The variance distributions for each type of cell are represented graphically in Figure 4 together with the *p* values. The *p* values show a net difference between nRBCs and cRBC^Plasma^, cRBC^HPL^ and cRBC^Plasma^, and a clear likeness between nRBCs and cRBC^HPL^. This observation holds both for static and oscillating traps. Interestingly, the *p* value between nRBCs and cRBC^HPL^ decreases for the oscillating trap, indicating a different behavior of the cells in movement. The variance for nRBCs is higher, suggesting that the cell membrane of nRBCs fluctuates more than cRBC^HPL^ when the cell moves in fluid.

To investigate the contribution of different frequencies to the fluctuation signal, we calculated the power spectrum density (PSD) for each cell and calculated the mean PSD corresponding to a cell type. The PSDs for the three types of cells in a fixed trap are presented in Figure 5, showing that nRBCs and cRBC^HPL^ have similar behavior and values for frequencies f < 500 Hz. The PSD values for cRBC^Plasma^ are lower and decay faster, indicating a reduced membrane fluctuation contribution in this range of frequencies with respect to nRBCs and cRBC^HPL^. The PSD decay follows a power law: PSD~f ^a^, with the values of the power coefficient a, is in the range: [−1.93, −1.83] for frequencies f < 300 Hz. These values are in good agreement with the theoretical prediction: PSD~f^−1.67^, calculated in the limit of a flat membrane and local measurement of the RBC edge fluctuations [24]. When the cells are moving in the oscillating trap, the distance between the PSD curves increases (data not shown), indicating that under dynamic conditions, nRBCs and cRBC^HPL^ preserve the CMFs better than the cRBC^Plasma^.

### 3.3. Morphological Parameters of Native vs. Ex Vivo-Generated RBCs Measured by DHM

Using the DHM method and morphological parameter definition described in Section 2, we measured and calculated the morphological parameters for the three types of cells considered in our study: nRBCs, cRBC^Plasma^ and cRBC^HPL^. The results are illustrated in Table 2.

The smallest projected area corresponded to the cRBC^Plasma^ (CA = 41.05 µm^2^), which also had the highest sphericity coefficient, CS = 1.04, and second highest volume, CV = 125.5 µm^3^ (or fL). Although the projected CA is small, the CV for cRBC^Plasma^ is big because of the cell shape. In fact, the shape of these cells is predominantly spherical–convex, different from the biconcave shape of the nRBCs. Thus, the mean height for cRBC^Plasma^ is also maximum among all the cell types (hm = 3.06 µm), confirming this observation. The sphericity of cRBC^Plasma^ cells was also confirmed by the optical tweezer experiments, where the cRBC^Plasma^ was trapped without being deformed and maintained its spherical symmetry (while nRBCs are clearly deformed by the OT forces, denoting that they are more deformable, see Section 3.1 and Section 3.2).

The biggest projected CA was found for the cRBC^HPL^ (CA = 70 µm^2^) though it is not associated with the biggest volume (CV = 107, 1 < 133.6 µm^3^). This is due to the biconcave shape of the cell, as also reflected by the sphericity coefficient (CS = 0.67). From a morphological point of view for parameters, CV, CS, MCH and hm have similar values for nRBCs and cRBC^HPL^. The difference in CA tells us that cRBC^HPL^ are slightly more flattened than nRBCs.

We also found that the mean corpuscular hemoglobin (MCH) values of cRBC^HPL^ and nRBCs are similar and confirmed the MCH values reported for nRBCs elsewhere [30]. Furthermore, these findings match well with the published reference values. For nRBCs, CV is in the range: (80.195.3) fl compared to 95.2 ± 16.6 fl and MCH = [27–33.2] pg compared to 25.24 ± 5 pg measured by DHM [38]. For cRBC^Plasma^, CV = 141.5 ± 9.7 fl compared to 125.5 ± 43.3 fl and MCH = 35.9 ± 7.6 pg versus 31.17 ± 11.7 pg. For cRBCs, lipid supplementation reference values of CV = 128 ± 11.7 fl and MCH = 32.7 ± 1.3 pg are slightly higher than the values from DHM analyses [32].

From Table 2, we notice that the standard deviations for CA and CV are bigger for cRBC^HPL^ and cRBC^Plasma^ than for nRBCs, meaning that the projected area and the volume are more spread for the generated RBCs. However, the standard deviation for cell sphericity, CS, is similar for all three types of cells and confirms that generated cRBC^HPL^ have a biconcave shape close to that of nRBCs.

### 3.4. Cell Membrane Fluctuations of Native vs. Ex Vivo-Generated RBCs Measured by DHM

We measured and calculated the cell membrane fluctuation CMFs by using the DHM methodology described in the Materials and Methods. To check that the cell fluctuations are correctly identified and separated from the background noise, we first measured the optical phase difference (OPD) for an nRBC cell and evaluated the OPD fluctuations for three randomly chosen pixels: on the cell ring, cell dimple and background, as shown in Figure 6a. The OPD for each pixel fluctuates around a mean phase value, which is higher for the cell (ring > dimple) and lower for the background (Figure 6b). Calculating the standard deviation (STD) of OPD fluctuations at the three pixels, we found that the STDs corresponding to the ring and dimple are similar (0.053 and 0.051) and, importantly, considerably higher (about two times) than the background (0.028). Together with the mean value, this shows that the signal is considerably higher than the background noise, allowing us to reliably measure the cell membrane fluctuations.

To confirm the validity on the OPD fluctuation measurements, we calculated the height fluctuations at each pixel of the cell, using Formula (3). The standard deviation STD_pix of the height fluctuations was then calculated for each pixel. The result for an nRBC is shown in Figure 7. STD_pix varies over different regions of the cell, with values ranging from 10 to 28 nm. To evaluate the CMFs for the whole cell in a single value, we calculated the mean value of STD_pix over all the pixels within the cell (Equation (9)). A high CMF value is associated with high membrane flickering/flexibility.

The results obtained for the three types of cells are presented in Table 3. The CMF amplitude of the nRBCs is the highest, confirming the results obtained by OOTs. However, the CMF values for cRBC^Plasma^ and cRBC^HPL^ show an inverted trend. Trying to explain this difference, we analyzed how CMF correlates with the morphological factors, following a quantitative analysis previously reported in [30]. We found that the CMF exhibits a negative correlation with the sphericity coefficient CS projected CA and CV for nRBCs and cRBC^HPL^, revealing results which are in line with the CMF analysis of stored RBCs [30]. The CMF amplitude of the cRBC^Plasma^ does not follow this trend.

Although we do not have a clear explanation for this difference, we think it might be related to subtle dissimilarities of the cytoplasm composition for different types of cells, and the fact that in DHM the cell height is derived from the phase retardation of the light passing through the cell. Thus, we observed that if we correct the measured CMF by the height normalization coefficient, c = hm_RBC_Type/hm_nRBC, the CMF values are rearranged in a sequence following the expected trend. These corrected values are also in agreement with the measurement results using OOTs, indicating that nRBCs are the most flexible and cRBC^Plasma^ the least, with cRBC^HPL^ in between.

## 4. Discussion

### 4.1. OOTs Allow the Investigation of Cell Deformation under Forces of Similar Strength as Those on Cells in the Blood Stream

In this work, we demonstrate OOTs as a useful and versatile technique to probe the biomechanics of native vs. ex vivo-generated RBCs. In OOTs, the radiation pressure of light interacts with the cell, inducing cell trapping and deformation without a mechanical contact. Optical trapping allows for studying the cell suspended in solution, away from the wall surfaces of the experimental chamber. The radiation pressure of light has a clepsydra-like shape, following the intensity’s spatial distribution, with the highest pressure at the clepsydra neck (focus of the laser beam) [37]:RP = n_m_ × P/A/c(10)
where n_m_ is the refractive index of the medium, A is the laser spot area and c is the speed of light in vacuum. Thus, if we consider a laser beam of power P = 20 mW focused through a high-numerical aperture lens (NA = 1.25) onto a focal spot of diameter d~0.6 µm, in a liquid chamber (n_m_~1.33), the radiation pressure in the focus is RPmax~300 Pa. Due to the light beam focusing through the high-NA lens, the radiation pressure rapidly drops along the optical axis, before and after the focus. For the above numerical values, which correspond to our OOT experimental setup, they result in an attenuation factor of ~19 × L^2^ for the radiation pressure, where L is the distance from the focus, expressed in microns. Considering the average diameter of an RBC is about 8 µm, the magnitude of the radiation pressure at the extremities of cell in the optical trap (L = ± 4 µm) drops about 300 times to RP_L = 1 Pa. The spatial distribution of the radiation pressure is symmetric and well controlled by the optical configuration. Therefore, the interaction between light and cell manifests in forces, which for a given value of the radiation pressure, depend only on the local material and morphological properties of the cell. A rough idea of the strength of forces locally acting on the cell is given by:F = Q × RP × Ac(11)
where Ac is the light–cell local contact area and Q is a dimensionless coefficient, which considers the material and morphological properties of the cell [36,37]. The value of Q ranges from 0.03 to about 0.3 for cells (Qmax = 2 for perfectly reflecting objects), meaning that for a contact area Ac =1 µm^2^ the forces acting on the cell range from less than 1 pN to 90 pN. The spatial distribution of these forces over the cell is neither homogeneous nor isotropic and hence they can induce cell folding and buckling.

Cell buckling at the entry of a blood capillary is preferred to cell shearing, because the required bending energy is lower than the shear energy or compression [16,39]. The configuration of the OOTs is ideal to mimic the passage of RBCs through capillaries. The pressure exerted on the erythrocytes in capillaries is of the order of the pressure exerted with OOTs [40,41]. Moreover, the oscillatory displacement of the cell represents an additional feature enabling us to study the cell flowing in liquid for a determined number of cycles. Although, here, the traveling distance is limited to 7 µm, a longer distance of tens of microns could be implemented, shaping the trapping laser into a Bessel beam [42].

### 4.2. The Bending Modulus Can Be Estimated from the Cell Deformation and Membrane Fluctuations

By tuning the power of the trapping laser from P = 20 to 30 mW, we always observed buckling for nRBCs and cRBC^HPL^ and never for cRBC^Plasma^. Reducing the power to P = 15 mW, about 60% of nRBCs and cRBC^HPL^ folded, and hence we settled on this value as the minimum power for nRBC and cRBC^HPL^ buckling. The corresponding radiation pressure at the neck is RP~225 Pa. Interestingly, this value is in the range of the pressure values applied to induce RBC buckling in micropipette experiments [39,40]. The experimental approach is also similar, with the difference that the radiation pressure of light is used in OOTs instead of the fluid aspiration pressure in a micropipette. Thus, the bending modulus can be derived from the pressure value at which the cell buckles, using an analogous formula:B~k × d^3^ × P/8(12)
where k is the ratio between the diameter of the laser beam at the focus (clepsydra neck) to the cell outer radius, d is the pipette inner diameter and P is the radiation pressure at the neck. In the micropipette suction technique, k is the ratio between the inner diameter of the pipette and the diameter of the cell, d is the inner diameter of the micropipette and P is the aspiration pressure. With the experimental values specified above for cell buckling in our OOT experiments (P~225 Pa, d = 0.6 µm, k = 0.075), the bending modulus is B~3.6 × 10^−19^ J, a value well in line with the results obtained from pipette experiments [39,40].

On the other hand, the variance of the membrane fluctuation has been used in local measurements of RBCs to establish an approximate formula for the bending modulus, B [24]:B~(6 × 10^−3^) × K_B_ × T × R^2^/V (13)
where K_B_ is the Boltzmann constant, T is the temperature, R is the cell radius and V is the variance of the membrane fluctuation. From this approximation, we see that the bending modulus is proportional to R^2^/V, i.e., for the same cell radius, a higher variance V of the membrane fluctuation is related to a lower value of the bending stiffness B. The variance measured in OOTs is expressed in mV^2^ and cannot be converted to length units. Therefore, it cannot be directly used to calculate the bending modulus, but it can be employed to compare different types of cells from the bending rigidity point of view if the radius is defined. The cell radius can be measured from the cell image in OOTs or from the hologram reconstruction in DHM. DHM provides the variance of the cell membrane fluctuations in nm^2^. Using the cell radius and the variance determined from the DHM experiments in the equation above, we obtain values of the bending modulus in the range 1−4 × 10^−19^ J. Although these values are in the range of bending rigidity values reported in the literature [26,39], confirming our experiments, care should be taken to assign absolute values to compare different types of cells. The cell radius value is a determinant in calculating the bending modulus, but the cells are not spherical and hence it is difficult to find a well-defined criterion for the radius which corresponds to all the cell types.

### 4.3. DHM Provides Information Complementary to OOTs, Allowing a Faithful Identification of nRBCs vs. cRBCs

Due to its ability to reconstruct the 3D profile of the cell, DHM provides useful information on the cell morphology, complementary to the OOT approach. The cell volume, CV, and the sphericity coefficient, CS, are two important parameters to assess the morphologic similarity between native and generated erythrocytes, as shown in Section 3.3. CV and CS found for nRBCs in our DHM approach are confirmed by the values reported elsewhere [30], while the application of DHM to generated cRBCs is, to our knowledge, new. Investigation of the morphology of cRBCs was reported in the literature using other methods like atomic force microscopy (AFM) [43] and scanning electron microscopy (SEM). For both methodologies, fixation is crucial to obtain high-resolution images. Glutaraldehyde fixation (1–2%) is widely used and commonly reported not to alter the cellular topographic of native RBCs, but this cannot be not clearly stated for cRBCs. In a recent study, for fixations of cRBCs with glutaraldehyde concentrations over 0.5%, artefacts in shape were observed [32]. Even immobilization of the cells with poly-L-Lysine alone increases rigidity and alters the shape of the cells [44]. Hence, technologies that do not need fixation or immobilization of target cells, like OTs and DHM, are beneficial for unbiased measurements.

Calculating the dry mass of the cell DHM allows us to evaluate the mean corpuscular hemoglobin (MCH) which is another important parameter for erythrocyte characterization. The DHM method is quite simple and fast: tens of cells situated in a field of view of about 0.2 × 0.2 mm can be measured simultaneously in several seconds.

Cell membrane fluctuations (CMFs) can be measured at different points of the cell and averaged to get the CMF value for the entire cell, as shown in Section 3.4. The result is expressed in terms of the standard deviation of the height fluctuations over all the cell pixels. The values obtained for nRBCs are in line with the results reported in the literature for nRBCs [30] and allow us to compare nRBCs to generated cRBCs. For nRBCs and cRBC^HPL^, the results obtained by DHM (Table 3) follow the same trend as the results obtained by OOTs (Table 1).

The complementarity of DHM to OOTs can be used for a multi-parameter analysis of cells in different conditions. DHM can be applied only to cells laid on a surface while OOTs allow us to study the behavior of the cells under forces comparable to those present in the blood stream. Both techniques provide useful information regarding the bending modulus and can be successfully used to characterize and compare different types of red blood cells.

### 4.4. Conclusions about RBC Properties Based on DHM and OOT Data

Native erythrocytes have a lifespan of 120 days in the circulation. During this period, they undergo numerous reversible deformations but keep their structural integrity. Three features are most important for this: the cellular surface area to volume ratio; the cytoplasmic viscosity and the membrane deformability [45]. This typical RBC deformation potential is based on membrane and cytoskeletal characteristics.

In the present study, cRBC^HPL^ showed high similarity to nRBCs both in morphology as well as in flexibility under optical forces, whereas cRBC^Plasma^ displayed a more spherical, round shape with lower flexibility. The lower energy of bending in cRBC^HPL^ compared to nRBCs might be due to differences in membrane curvature. This in turn is supposed to depend on the cellular lipid content [46]. The cell-enveloping plasma membrane consists of cholesterol and phospholipids in equal amounts [47]. The four main phospholipids, phosphatidylethanolamine (PE), phosphatidylserine (PS), phosphatidylcholine (PC) and phosphatidylinositol (PI), are distributed asymmetrically between the two leaflets of the lipid bilayer, while cholesterol is equally distributed. In a recent publication, we described culture-related lipid malnutrition of cRBCs without additional lipid supplementation (cRBC^Plasma^). cRBC^Plasma^ revealed a mean cholesterol content of 23.83 ± 10.31%, compared to 49.27 ± 9.87% in nRBCs [32]. Consequently, these cells showed different biomechanical properties from native cells in ektacytometry and osmotic resistance analysis. Cellular imaging techniques confirmed differences in cell geometry, as the majority of cRBC^Plasma^ showed an increased cell volume and a more spherical shape, as well as a higher grade of vesiculation. cRBCs supplemented with HPL revealed a more discoid shape and a deformation potential close to that of nRBCs (unpublished data from our group). In the current work, cRBC^HPL^ showed morphological parameters, but also flexibility and cell folding, comparable to nRBCs. In contrast, cRBC^Plasma^ displayed higher rigidity, which is supposed to be a consequence of their sphericity. The amount of membrane cholesterol may play a crucial role in the different behavior (unpublished data of our group). In vivo, hereditary RBC disorders like spherocytosis, elliptocytosis, sickle cell disease or thalassemia, all showing characteristic RBC shapes, are also characterized by decreased deformability of the cells [48,49,50]. That suggests there must be additional key players besides membrane cholesterol. As stated by Mohandas et al., excess surface area to volume ratio is further crucial for cellular deformation [51]. This is due to the maintenance of cohesion between the cytoskeleton and the lipid bilayer, which avoids vesiculation [52]. cRBCs lack terminal cytoskeletal remodeling to biconcavity ex vivo, therefore an impaired, immature functionality of their cytoskeleton is assumed to additionally impact deformability. This is investigated in ex vivo differentiation of RBCs in an ongoing study of our group.

Publications on nRBCs under healthy and pathological conditions use both DHM and OTs [23,30]. Data on cultured RBCs using OTs are scarce and from mainly bead-based methodologies. Like the well-established technologies AFM and SEM, these methods suffer from low throughput. The newly implemented bead-free and simpler OT approach enables faster cell manipulation. DHM analyses allow the observation of thousands of events per minute and are ideal for the global characterization of RBCs. Although there exist reports on DHM and native or cultured red blood cells [28,31,53], to the best of our knowledge, we are the first to do a systematic comparative study on both cRBCs and nRBCs.

The DHM and OOT approaches presented here reveal a high level of information on small differences between the different RBC types, investigating deformation and elasticity under fields of forces similar to those impacting RBCs in microcirculation. With a higher throughput under physiological conditions, these new methodologies might be of interest not only for quality control of ex vivo-generated RBCs, but also for the control of storage lesions of RBCs in blood banks and diagnostics of inherited RBC disorders.

## Figures and Tables

**Figure 1 cells-10-00552-f001:**
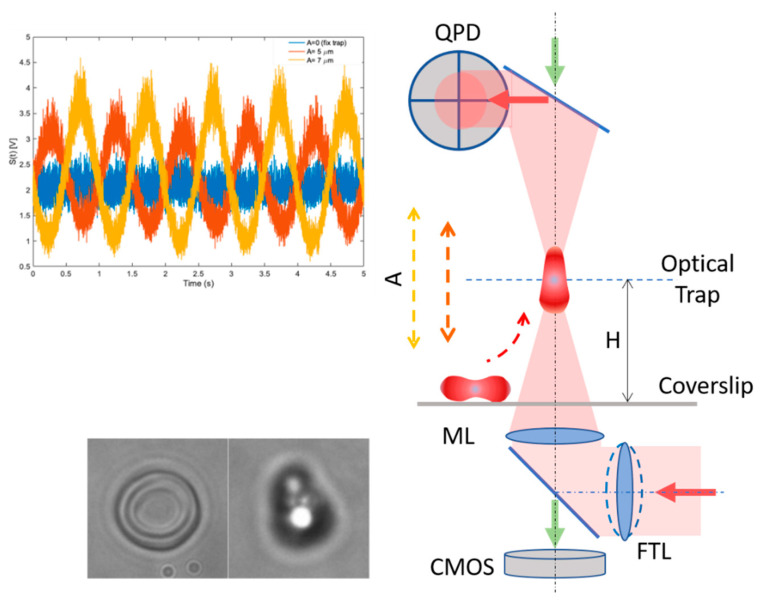
Schematic of the oscillatory optical tweezer (OOT) setup. The red blood cell (RBC) is picked up from the coverslip into the optical trap by the trapping IR laser (red arrows). Cell imaging (bottom left inset) is done by the microscope lens (ML) on the sensor of a CMOS camera by up–down illumination (green arrows) while the intensity fluctuations of the light scattered by the cell are recorded on the Quadrant Photo Detector QPD (top left inset). Using the focus tunable lens (FTL), the position of the optical trap is moved vertically in an oscillatory motion (lens with dotted line illustrates the change of the FTL curvature).

**Figure 2 cells-10-00552-f002:**
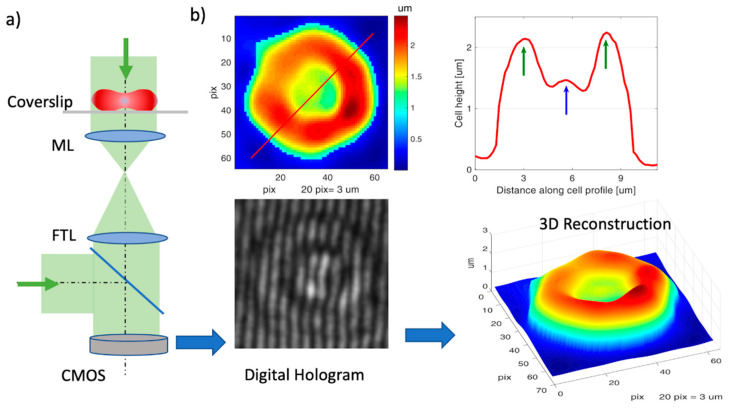
Digital holographic microscopy (DHM) for 3D RBC imaging. (**a**) Schematic of the DHM setup. Green arrows: laser beam; ML: microscope lens, TL: tube lens, CMOS: camera sensor; (**b**) recorded digital hologram on CMOS and 3D reconstruction (bottom); height profile of the cell (top right) along the red line shown in reconstruction (top left).

**Figure 3 cells-10-00552-f003:**
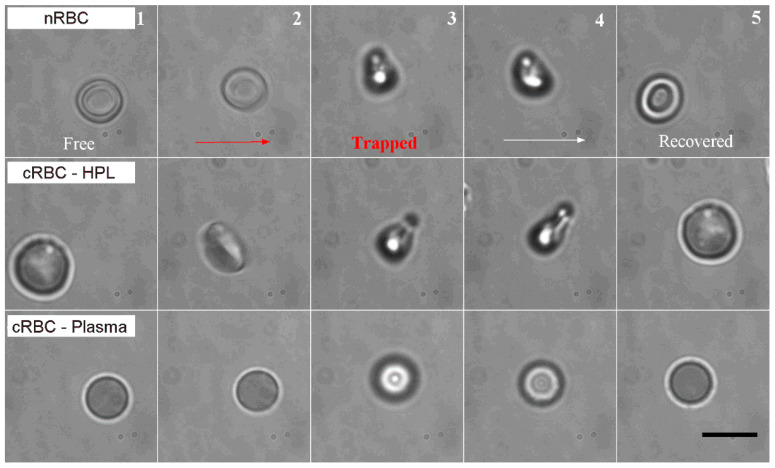
Deformation of nRBCs and cRBC^HPL^ as well as cRBC^Plasma^ during optical trapping. (1) Cell laid on the substrate; (2) cell begins to be attracted to the trap; (3) cell folded in a stable shape during trapping; (4) cell just released from the trap, laser trap off; (5) cell recovering its shape (biconcave shape). Scale bar: 10 µm. HPL = human platelet lysate

**Figure 4 cells-10-00552-f004:**
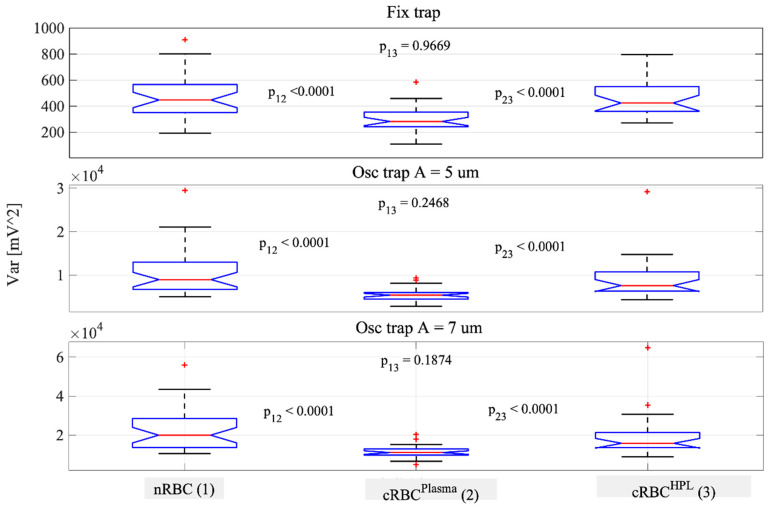
Variance results for nRBCs, cRBC^Plasma^ and cRBC^HPL^ for fixed and oscillating traps: A = 5 µm and A = 7 µm, respectively. The distributions of the variance values for each cell type are represented graphically using the boxplot method. The *p* values are calculated with a Mann–Whitney U test.

**Figure 5 cells-10-00552-f005:**
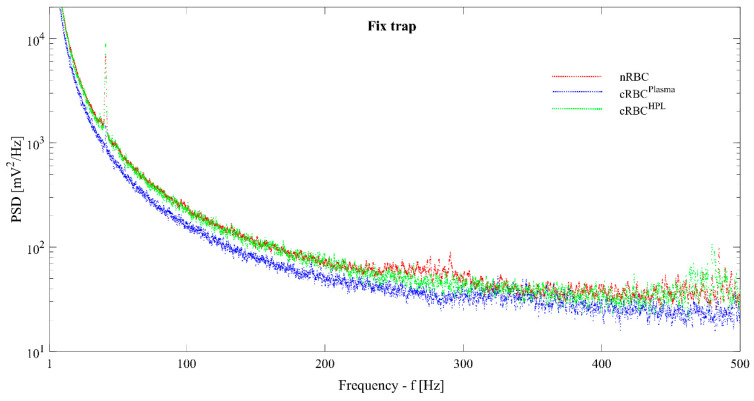
Power spectrum density (PSD) function calculated for fixed trap. Each PSD curve is obtained averaging over all the PSDs of the cells of the same type. The peak at the frequency f = 42 Hz is not specific to the cells; it is an artefact due to an external frequency and it can be ignored.

**Figure 6 cells-10-00552-f006:**
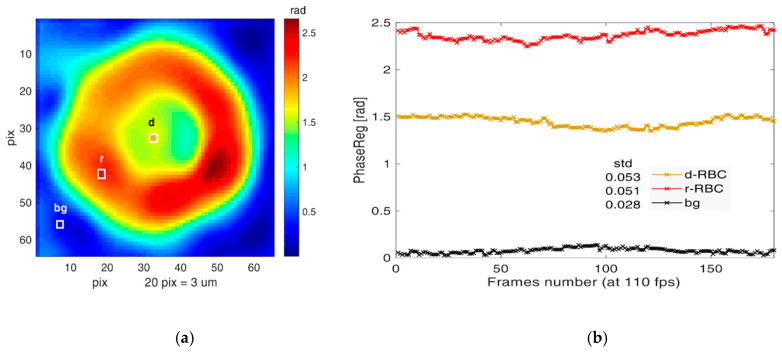
(**a**) Optical phase difference (OPD) over the cell, with three points chosen randomly: ring, dimple, background; (**b**) OPD fluctuations recorded for t > 1 s and standard deviation calculated.

**Figure 7 cells-10-00552-f007:**
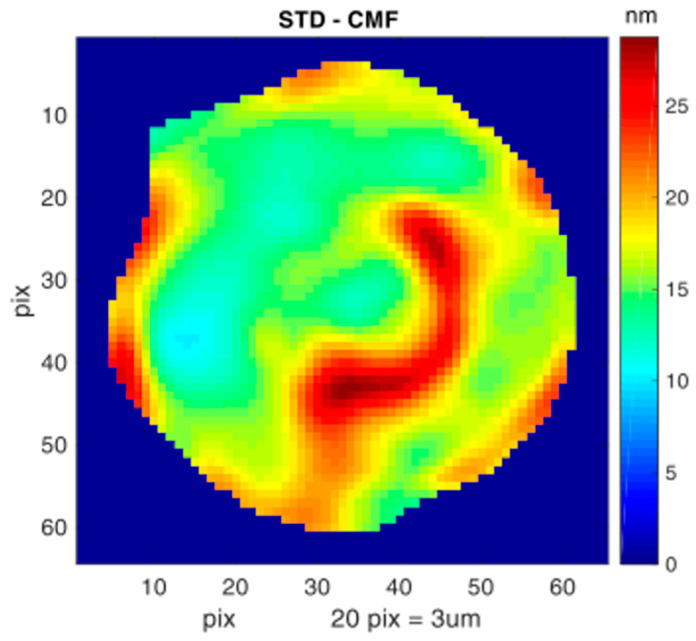
STD_pix—the standard deviation distribution over the cell area.

**Table 1 cells-10-00552-t001:** Variance for cells in fixed and oscillating traps.

**Variance** Mean/Std (mV^2^)	**nRBC** (*n* = 32)	**cRBC^Plasma^** (*n* = 24)	**cRBC^HPL^** (*n* = 30)
Fixed trap	460/160	300/100	470/140
Osc. A = 5 µm	10,800/5500	5600/1500	9200/5100
Osc. A = 7 µm	22,600/10,800	11,300/3100	19,900/11,600

**Table 2 cells-10-00552-t002:** Cell morphology parameters. **CA**: Cell area; **CV**: Cell volume; **CS**: Cell sphericity coefficient; **MCH**: Mean corpuscular hemoglobin; **hm**: Equivalent cell height.

Morphology		CA	CV	CS	MCH	hm
		mean ± std	mean ± std	mean ± std	mean ± std	mean
Cells	*n*	µm^2^	µm^3^ (or fL)	-	Picogram (pg)	µm
**nRBC**	25	55.42 ± 9.2	95.2 ± 16.6	0.57 ± 0.1	25.24 ± 5	1.72 ± 0.4
**cRBC^Plasma^**	24	41.05 ± 14.4	125.5 ± 43.3	1.04 ± 0.1	31.17 ± 11,7	3.06 ± 0.6
**cRBC^HPL^**	29	70 ± 21.7	107.1 ± 37.8	0.671 ± 0.4	28.1 ± 10.9	1.53 ± 0.3

**Table 3 cells-10-00552-t003:** Cell membrane fluctuation (CMF) determined by DHM. The height normalization coefficient, c, is calculated using the mean height, hm, given in Table 2. Corrected CMF is obtained from measured CMF divided by the normalization coefficient c.

CMF		Cell Measured		Background		Height Normalization	Cell Corrected
		mean	std	mean	std	c	mean
cells	*n*	nm	nm	nm	nm	-	nm
**nRBC**	25	54.4	10.8	24.7	6.2	1.00	54.47
**cRBC^Plasma^**	24	53.8	21.4	25.3	9.1	1.78	30.27
**cRBC^HPL^**	29	37.9	14.3	19.4	7.9	0.89	42.61

## Data Availability

Data can be made available upon request to the corresponding authors.

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
