# Peer review of "Biomechanics of Ex Vivo-Generated Red Blood Cells Investigated by Optical Tweezers and Digital Holographic Microscopy"

_cells, 2021, doi:10.3390/cells10030552_

Round 1

Reviewer 1 Report

In regards to manuscript cells-1056395 "Biomechanics of ex vivo generated red blood cells investigated by optical tweezers and digital holographic microscopy" the authors present an optical methodology to assess biomechanical properties of natural red blood cells (nRBC) and two types on in vitro obtained RBC: plasma RBC (pRBC) or human platelet lysate RBC (hplRBC). The optical methodology consists on two techniques: optical tweezers (OT) and image analysis through digital holographic microscopy (DHM). The three types of RBC are analysed and compared to assess their biomechanical properties in order to propose artificially in vitro obtained RBC for clinical use.

The manuscript is well written, clearly exposed and the results are adequate. Through the use of both static and dynamical tests employing OT and DHM, the authors provide experimental proof that hplRBC are adequate surrogates for nRBC, as far as biomechanical properties are conceerned. On the other hand, pRBC seem less promising in this sense. The optical techniques employed can have many other interesting applications in the field of cell biomechanics.

There are a few minor issues to deal with before the manuscript is ready for publication, as indicated below.

In section 2.2 it would be desirable to provide more data on the laser system employed to implement the OT. For example, power range, if the laser is CW or pulsed, irradiance level at the focal plane. The same can be said in section 2.3 for the laser employed in DHM (wavelength, exposure time, etc.). In the case of cell membrane fluctuations, please indicate if the laser beam was focused or not to obtain the interference pattern.

Some relevant laser information (power) is provided in the Discussion (see section 4.1). This information should be presented first in the M&M section.

At the end of the penultimate paragraph of the Introduction it reads "DHM requires and additional time..." and should read "DHM requires an additional time...." Please, correct.

Author Response

We thank the reviewer for the comments. We added more data in section 2.2, 2.3, and 4.1 as requested.

The changes in the text are marked in red.

The typing error in line 97 was corrected from "and" to "an".

Reviewer 2 Report

In this manuscript, cultured RBC's and natural RBC's are compared by optical trap investigations of the membrane and cell
elasticity on one side (OT), including cell membrane fluctuations (CMF) and by digital holographic
microscopy (DHM) to investigate 3D morphology and dynamics.

It is argued that for qualitative studies by optical trapping methodology, direct trapping of the  individual cells is preferred for investigations in which microspheres attached to the RBC's are
trapped. The benefit of the latter is the fact that the optical force can be known quantitatively.
The argument of the paper seems to be that the direct trapping is to prefer to allow for good statistics / reasonable throughput, an argument that I only partially follow - do the authors
not do single cell experiments, after all?
Further, the argument is challenged by the desire to do quantitative analysis anyway, as presented in the results and discussion section.

The statistics behind the results presented is limited (n=32 at max, in one case, more "typical" numbers are n~25), but appropriately discussed.

Results for the variances of cell membrane fluctuations (CMF) are listed in Table 1, however, one misses information on the unit of the numbers stated. The p-values listed in Fig. 4 are also unclear.

From the optical trapping data, exponents are extracted from PSD fits, in three frequency intervals. It is unclear why this division in
different frequency ranges is introduced, and any effect of the optical trap is unclear. This point should be further discussed.

Table 2: Assumes um is micro-meter and pg is pico-gram.

Table 3 / discussion at the bottom of p. 12: I am not sure I understand the need for the renormalization, or why the renormalized/corrected CMF values are "more related to the reality" (statement 3 lines from bottom of p. 12). One may argue that the renormalized CMF's are the values to compare, but what is measured is "the reality", not the renormalized number.

Just above and below Eq. (12), the authors claim a correspondence between the radiation pressure and the suction pressure in a pipetting experiment. The comparison is highly elusive and if the authors wish to keep the discussion, a clearer physical explanation is needed.

In general, the steps taken to seek to get quantitative results are in contrast to the initial arguments for the methodology applied in this paper rather than bead-based methodology.
I rather suggest that the authors investigate if a multi-parameter analysis could be used to document similarity and differences between the cultured and the natural RBC's ?

In general, the paper is well written, however, the paragraphs around Eq. (13) contain almost incomprehensible sentences. I encourage the authors to work on the writing and be clear on whether they believe that they can do quantitative comparisons, or qualitative comparisons only.

In conclusion, the manuscript presents well described experiments that seem to be carried out with great care, but an analysis and discussion that still requires some work before the paper is publishable.

Author Response

We thank the reviewer for useful comments, which helped us to improve our manuscript. Some text was removed and other was re-formulated or added, as marked in red. For answers to the comments please see the attachment. 

Reviewer 3 Report

This is a very well presented work.

I have no criticism to comment on.

Author Response

Many thanks for this utmost positive review, we highly appreciate this.

Round 2

Reviewer 2 Report

The authors responded well to most of my previous comments. While formulations may still benefit from a careful rereading and rephrasing, the modifications have overall benefitted the manuscript.

One comment still prevails, though: I still think the data analysis could be sharper. What do we really learn from the PSD's? And since the authors agree that a multi-parameter (statistical) analysis is feasible, why don't they demonstrate it?

Author Response

Please find the answers in the attached file.
